# Organic Carbon, Nitrogen Accumulation and Nitrogen Leaching as Affected by Legume Crop Residues on Sandy Loam in the Eastern Baltic Region

**DOI:** 10.3390/plants12132478

**Published:** 2023-06-28

**Authors:** Liudmila Tripolskaja, Asta Kazlauskaite-Jadzevice, Almantas Razukas

**Affiliations:** Lithuanian Research Centre for Agriculture and Forestry, Vokė Branch of the Institute of Agriculture, Zalioji Sq. 2, LT-02232 Vilnius, Lithuania; liudmila.tripolskaja@lammc.lt (L.T.); almantas.razukas@lammc.lt (A.R.)

**Keywords:** nitrogen, organic carbon, leaching, above-ground biomass, below-ground biomass

## Abstract

Legumes have a wide range of positive effects on soil properties, including nitrogen and carbon storage, soil structure and the phytosanitary condition of crops. From an agronomic point of view, legumes are most valued for their ability to take up atmospheric nitrogen in symbiosis with nitrogen-fixing bacteria. The aim of this research was to determine the effect of legume residues (peas, fodder beans, narrow-leaved lupins) on the N (N_total_) and organic carbon (C_org_) accumulation in soil and N leaching under temperate climate conditions. The experiment was carried out in lysimetric equipment in 2016–2023. The effect of legumes on C_org_ and N_total_ accumulation in soil and N leaching were studied in a *Fabaceae–Cereals* sequence. Three species of legumes—peas, fodder beans and narrow-leaved lupines—were tested; spring barley (*Hordeum vulgare* L.) was grown as a control treatment. The lysimeter surface area was 1.75 m^2^ and the experimental soil layer was 0.60 m (sand loam Haplic Luvisol). It was found that after harvesting, more residues were incorporated into the soil with lupines (*p* < 0.05), which, compared to pea and bean residues, increased N_total_ and C_org_ concentrations in the soil. There was a strong correlation (r = 0.95) between the N_total_ concentration in the soil and the N amount incorporated with residues. Mineral N released during residue decomposition was leached from the humic horizon under conditions of excess moisture in the autumn–winter period and increased the nitrate concentration in the lysimeter water. The increase in concentration was recorded within 5 to 6 months after the application of the residues. As a result, the N leaching losses increased on average by 24.7–33.2% (*p* < 0.05) during the year of legume cultivation. In the following year, after legume residue incorporation, the effect of residues on nitrate concentration and N leaching decreased and did not differ significantly from that of barley residues.

## 1. Introduction

The rotation of legumes is important for sustainable functioning of the agroecosystem, as they have a wide range of beneficial effects on soil properties, including nutrient concentrations and crop phytosanitary conditions. From an agronomic point of view, legumes are most valued for their ability to take up atmospheric N in symbiosis with nitrogen-fixing bacteria and use it to form biomass yields. Depending on the species and growing conditions, legumes can accumulate 15.0–20.0 g N m^−2^ of symbiotic N [1,2,3]. This reduces the use of synthetic N fertilisers for plant fertilisation and, consequently, the cost of growing crops [4,5]. Some legume species are also able to take up mineral phosphorus compounds that are difficult to dissolve in soil and incorporate them into their biological cycles [6,7]. In addition to these benefits, the harvesting of legumes also leaves a large amount of above- and below-ground residues in the field, which can supplement the C_org_ and N stock in soil after mineralisation. Compared to cereal crops, the chemical composition of legume residues (C/N ratio) is much more favourable for the formation of humic substances, which has a more significant effect on soil chemical and physical properties [8,9,10]. Increasing soil humus content improves soil structure, water-holding capacity and other soil physical properties. 

However, the decomposition of plant residues also releases mineral N and other mineral elements [11,12,13]. The dynamics of this process depend on climatic factors (temperature and humidity conditions), the chemical and biochemical composition of plant residues (C/N ratio, cellulose, lignin content), agrotechnical practices (residue chopping, depth of incorporation) and other factors [8,14,15,16,17]. In Central and Eastern Europe, the mineralisation of post-harvest crop residues takes place during the autumn–winter period, when part of agricultural land is not covered with overwintering crops. Therefore, mineralised N can be leached out of the humus soil layer during the autumn period in the presence of favourable air temperatures and high rainfall and increase nitrate concentrations in groundwater [18,19,20]. Managing this process is of great importance for countries bordering the Baltic Sea, as elevated levels of nitrates in groundwater can negatively affect functioning of the marine ecosystem [21,22,23,24]. 

Plant residue destruction processes in soils and N leaching are also significantly influenced by climate change, the variation direction and magnitude of which depend on the specific deviations of rainfall and air temperature from the standard climatic normal (SCN) for a given region. In Lithuania, temperatures started to rise rapidly in the late 20th century. Compared to 1961–1990, the temperature SCN for 1991–2020 increased by +1.2 °C [25]. Increasing air temperatures reduced the cold period duration, when frost forms on the soil surface, and the decomposition of plant residues and infiltration of precipitation do not take place. Climate change models for various scenarios (RCP4.5 scenario and RCP8.5 scenario) predict that an increase in air temperature will accelerate mineralisation processes in soil, which will substantially increase nutrient leaching [26]. In addition to temperature, precipitation also has a significant effect on the leaching of chemical elements. Of all meteorological factors, precipitation has been found to have the greatest influence on the magnitude of leakage (the degree of influence approaching 23%), with a lesser effect from maximum temperature values (approaching 12%) and a minimal effect from minimum temperature values [27]. In Lithuania, the precipitation amount has not changed significantly over the last 30 years (1961–1990 SCN 675 mm, 1991–2020 SCN 695 mm), however, during recent decades there has been an increase in precipitation in the cold season and a decrease in the warm season [28]. The length of seasons has also changed. It has been found that the cold season length in Lithuania has shortened by 16 days, while the summer season has become 13 days longer [25]. The growing season lengthening changes the duration of plant development and maturation, and influences microbiological processes in soil, such as soil C_org_, mineralisation and humification of plant residues as well as the leaching of mineralised compounds into the subsurface horizons. According to [29], potential N losses through the mineralization of N-rich residues from legumes after harvesting until the end of the growing season might be underestimated. Studies on the effect of legume residues on soil N cycling will help to better assess their impact on the N_total_ and C_org_ accumulation in soil and N leaching in the East Baltic region climate zone. 

The aim of this work was to evaluate the effects of the above-ground and below-ground parts of legume (pea, fodder bean, narrow-leaved lupine) residues incorporated in the soil after crop harvesting on the N_total_ and C_org_ accumulation in the soil and nitrate leaching under warming climate conditions in the temperate climate zone.

## 2. Results

### 2.1. Plant Biomass Yield

In Lithuania, legume crops occupy about 10% (134,400 ha) of arable land. Fodder beans and peas form the main part of the crops (in 2021 beans amounted to 77,700 ha and in 2022 56,000 ha; in 2021 peas amounted to 64,000 ha and in 2022 74,000 ha) (Official statistics portal of Lithuania). Narrow-leaved lupins are mostly used as siderates to improve soil properties. They are widely grown in acidic soils. Therefore, the mentioned plants were selected for this research.

The aim of this research was to evaluate the post-harvest biomass of fodder beans, peas and lupines for their impact on soil properties and nitrogen leaching. Estimating plant biomass and its qualitative differences provides an opportunity to justify differences in their impact on changes in nitrogen and carbon content in the soil and to justify differences in nitrogen leaching when growing one or another legume.

When legumes (peas, beans, narrow-leaved lupines) were grown for grain on low-fertility soils, narrow-leaved lupine had a greater ability to exploit its natural potential for overall phytomass formation. Its biomass at full maturity averaged 690.3 g m^−2^ (*p* < 0.05), while the beans and peas had slightly lower biomass (*p* > 0.05), and the barley had a biomass yield almost twice that of the lupine (Table 1). The grain yields were not proportional to the total plant biomass. When grown without N fertiliser, the beans produced the highest grain yield on sandy loam soil, with an average yield of 274.8 g m^−2^ (*p* < 0.05). Compared to the beans, the peas produced a significantly lower yield (197.8 g m^−2^, *p* < 0.05), while the lupines and barley had similar grain yields of 146.1–157.4 g m^−2^. Of all the plants tested, the lupines and peas formed more abundant above-ground biomass (419.9–478.3 g m^−2^), which remained on the soil surface after harvesting and after mineralisation complemented the soil with N and C_org_. 

The below-ground biomass of plants was significantly lower compared to the above-ground part. The pea roots were the least abundant in the soil after harvest, with only 20.3 g DM m^−2^. The barley and bean root biomass was twice that of peas (*p* < 0.05), while the lupine produced the highest root biomass compared to the barley and peas (65.9 g m^−2^, *p* < 0.05), and its root mass was very similar to that of the beans (*p* > 0.05). Looking at the total biomass of plant residues after harvesting, it can be confirmed that on sandy loam soil, the lupine crop had the highest residue content (544.1 g m^−2^, *p* < 0.05), the pea and bean crops had slightly lower residue contents and were not significantly different from that of lupine, and the barley crop had the lowest residue content (213.4 g m^−2^, *p* < 0.05). 

### 2.2. Concentration and Storage of Nitrogen and Carbon in Plant Biomass

Of all plant parts, the most N was found in grain. Among the legumes, the highest concentrations were found in the lupine and bean grains (*p* < 0.05), while peas had about 10 g N kg^−1^ less (Table 2). In terms of the above-ground part of the plant, the N concentrations were higher in the bean stems (11.3 g N kg^−1^, *p* < 0.05), lower in the pea and lupine stems (8.4–9.0 g N kg^−1^, *p* > 0.05), but not significantly different from those in the barley straw and bean stems. In the roots, the N concentrations were similar to those in the above-ground biomass. Higher concentrations were found in the pea roots (17.6 g N kg^−1^, *p* < 0.05), while other plants did not show significant differences in N concentrations (*p* > 0.05). 

The carbon concentrations in the above-ground plant residues varied from 42.6 to 35.5% (Table 3). Significantly higher concentrations of C (*p* < 0.05) were found in the peas and lupines, while the lowest concentrations were found in the bean residues. Similar differences in carbon concentrations between the different legume species were also observed for the roots. The roots showed slightly higher concentrations of C compared to the above-ground parts and significantly higher (*p* < 0.05) concentrations of C were also found in the roots of peas and lupines, and bean roots demonstrated the lowest concentration. 

The different concentrations of N and C also resulted in different ratios between the above- and below-ground parts. The barley residues had the highest C/N ratios in the above- and below-ground parts, at 76.4 and 57.8, respectively. Due to the higher N concentration in the tissues of the above-ground part of the legumes, the C/N ratios were lower compared to that of barley, varying in the range of 31.4 to 49.8 in the above-ground biomass, and in the range of 25.5 to 43.6 in the below-ground biomass. Fodder beans had the lowest C/N ratio in above-ground biomass and peas in below-ground biomass. 

The N accumulation in the plants depended on two factors: the N concentration and the biomass of a particular morphological part of a plant. The highest N accumulation in sandy loam soil was found in lupine biomass (Figure 1). The fodder beans accumulated 33% less N (*p* < 0.05) than the lupines and 52.1% less (*p* < 0.05) than the peas. Calculation of symbiotic N accumulation in the biomass of the tested legumes [30] showed that their capacity to accumulate atmospheric N varied considerably in sandy loam soils. The fodder beans accumulated significantly more symbiotic N (10.19 g N m^−2^, *p* < 0.05), while the N fixation of lupines was 30.0% weaker than that of beans, and that of peas was even twice as weak. 

Most of the N in the plants (54.6–73.2%) was concentrated in their grains. The highest N content (13.77 g m^−2^, *p* < 0.05) was accumulated in the bean grains (Figure 2). The lupine and pea grains accumulated similar N contents, 6.19–8.49 g N m^−2^, which were significantly lower compared to that of the beans. The N accumulation in the above-ground part of legumes was twice as low (4.01–8.03 g N m^−2^) compared to that of the grain. The lupine biomass accumulated the highest N content (*p* < 0.05) as it produced significantly higher above-ground biomass compared to the peas and beans. 

The root mass of legumes is not large and, although the N concentration was similar to that of the above-ground part of plants, the N accumulation was only 0.49–0.80 g N m^−2^. This represents 1.2–3.0% of the N accumulation in the whole plant. Among the legumes tested, the nitrogen accumulation in the roots of peas was significantly lower (0.19 g N m^−2^, *p* < 0.05) compared to the beans and lupines, as their root biomass was significantly lower.

### 2.3. Variation of Total Nitrogen and Organic Carbon Concentrations in Soil

During the research period (March 2016–February 2023), legumes were grown four times in the lysimeters. The plant residues incorporated into the soil after harvesting had a positive effect on the C_org_ accumulation in the humus layer. At the start of the experiment, its concentration in the 0–25 cm layer was 1.18 g C_org_ kg^−1^. After seven years of the annual cultivation of barley (control) and incorporation of only low-nitrogen straw into the soil, the C_org_ content in the soil decreased significantly (−0.07 g C_org_ kg^−1^), while in the case of cultivation of legumes (barley/legume) every second year, its concentration increased to 1.26–1.44 g C_org_ kg^−1^ (Table 4). The C_org_ accumulation in soil was more pronounced in the fodder beans and lupines, and less pronounced in the peas. The legumes also increased the N accumulation in the soil. Compared to the barley crop, the N_total_ concentration in the 0–25 cm layer was 0.06–0.16 g N kg^−1^ higher. The N accumulation in the soil was significantly higher with lupines. 

### 2.4. Nitrogen Leaching

After legume harvesting, the above- and below-ground biomass incorporated into the soil averaged 4.24–8.824 g N m^−2^, which is 3.2–6.6 times higher than that of the barley residues. In the course of organic matter destruction, the mineral N released is leached out of the soil humus layer and can increase nitrate concentrations in groundwater. This is confirmed by the experimental results. It was found that during the autumn and winter period, the N concentration in the lysimeter water increased after the incorporation of the residues of all plant species. While the average N concentration in the lysimeter water was 3–12 mg N L^−1^ in the spring period and 2–10 mg N L^−1^ in the summer period, it increased up to 60–70 mg N L^−1^ in autumn and winter (Figure 3). Compared to the barley straw, the effect of legume residues on N concentrations in the lysimeter water during autumn–winter during the first year of the experiment (2016) was negligible. However, repeated cultivation of legumes in 2018, 2020 and 2022 showed a substantial increase in N concentrations 2–3 months after their incorporation. The increase was recorded until February–March. In spring, the effect of the legume residues on the N concentration decreased and was not significantly different from the N concentration in the infiltrate after barley residues.

The N leaching losses depended on both the N concentration and the amount of precipitation that percolated through the soil layer. It was found that the rainfall infiltration was significantly higher in the barley crop (+7.6–12.1%, *p* < 0.05) compared to the legume crop (Table 5). However, in the following year, when barley was grown after legumes, the differences in infiltration were already non-significant in all treatments (*p* > 0.05).

Despite higher rainfall infiltration in the barley crop, the N leaching losses in the legume crop were significantly higher (+24.7–33.2%, *p* < 0.05) compared to barley. Of all the legume species tested, the lupines had the highest N leaching rate; nevertheless, the differences with the peas and fodder beans were not significant (*p* > 0.05). In the following year after legume cultivation, the effect of the incorporated post-harvest residues on N leaching was already insignificant and not significantly different from N leaching in the barley crop. 

## 3. Discussion

In the medium-humidity climate and on sandy loam soils, the lupines had a higher total plant biomass; however, it was not significantly different from that of the fodder beans and peas (*p* > 0.05). The abundance of biomass formed did not affect grain yield formation. The legumes efficiently utilised the potential soil fertility, resulting in a 26–75% higher grain yield compared to the barley, with the exception of the lupines. An analysis of the stability of grain yields in northern Europe has shown that the stability of legume yields is higher than that of other summer broad-leaved crops and only slightly lower compared to cereals [31]. They must therefore be cultivated more widely in Europe, as grain legumes produce high quality protein for food and feed, and potentially contribute to sustainable cropping systems. 

After grain harvesting, legume residues are incorporated into soil with an average of 371.8–544.1 g m^−2^ DM, which enriches the soil with N and C_org_. Legume residues have a favourable C/N ratio for mineralisation, resulting in the formation of humic materials during destruction, which promotes carbon sequestration in the topsoil [32,33]. The rate of decomposition of legume residues has been found to vary depending on the type and part of the plant, due to their different chemical compositions. The above-ground part of the plant (shoot residues) mineralises faster, while the root residue decomposes more slowly and this has a positive effect in a crop rotation in the second year [15]. Ref. [34] found that the C/N ratio of plant residues was a good predictor for net N mineralization, while the lignin concentration was a good predictor for carbon mineralization from both roots and shoots. In our experiment, the N_total_ concentration in the soil increased by 0.6–1.6 g N kg^−1^ during the experimental period when grain legumes were grown every second year compared to the barley crop. Significantly more (*p* < 0.05) N_total_ was found in the soil where the lupines had been grown. The bean residues had a lower effect on the N concentration compared to lupines, while the pea residues, although increasing the N concentration in the soil, did not show any significant differences with the effect of the barley residues (*p* > 0.05). There was a strong correlation (r = 0.95) of the N concentration in the soil with the N content incorporated with crop residues. However, it should be noted that not only the N content in residues and C/N ratios determine biodegradation processes, but other properties such as % cellulose, % lignin and polyphenol concentrations have also been shown to influence stubble decomposition [35].

The legume residues also stimulated C_org_ accumulation in the soil. At the end of this experiment, the C_org_ concentration in the soil with previously grown legumes was 1.5–3.3 g C_org_ kg^−1^ higher compared to that of barley. Of the legume crops tested in the experiment, the incorporation of fodder bean residues increased the C_org_ content the most (*p* < 0.05), while the incorporation of pea residues increased the C_org_ content the least (*p* < 0.05). A weaker effect of peas on C_org_ accumulation has also been confirmed by the experiments of other researchers [36]. However, many authors point out that the introduction of legumes into crop rotation promotes C_org_ sequestration [9,37]. According to [38], the effect of legume residues on C_org_ accumulation can be 30% higher compared to other crops. However, such an effect is not found when legumes are grown infrequently in rotation. The effect of their residues on C_org_ accumulation in soil may be negligible [39]. 

On average, 6.19–13.76 g N m^−2^ was removed from the field with the legume crops. That represented 54.6–73.2% of the total biomass N content. The post-harvest residues returned an average of 4.24–8.824 g N m^−2^ N to the soil, which was gradually released from the plant mass and facilitated N migration to groundwater. Under Lithuanian climatic conditions, a leaching moisture regime develops in soil during the autumn and winter periods, and losses of N leaching from agricultural land can be significant [40]. In our experiment, the increase in N concentrations in the lysimeter water was pronounced 2–3 months after the incorporation of plant residues. Depending on hydrothermal conditions, the legume residues increased N-NO_3_ concentrations in the infiltrate by an average of 14–70% in the autumn and 14–86% in the winter seasons compared to the barley. The lupine residues had a more significant effect on N leaching compared to the pea and bean residues, with 76–108% more N incorporated into the soil than with the pea and bean residues. The effect of legume residues on N leaching has also been shown [41] to depend on tillage practices. Ploughing of residues activates microorganisms and increases N leaching. It should be noted that the effect of the legume residues on N leaching increased with repeated cultivations in crop rotation and was insignificant in the first year of growing (2016). Some 5–6 months after the application of plant residues, the differences in the N concentrations in the infiltrate between barley and legumes became small. This suggests that the mineralisation of legume residues in the soil was quite rapid and had an effect on nitrate concentrations 5–6 months after incorporation. In the following year after the incorporation of the legume residues, the nitrate concentration in the infiltrate was similar to that after the incorporation of barley residues (*p* > 0.05). Various microbiological studies on the destruction of residues confirm that, at low C/N ratios, their mineralisation occurs rapidly [42,43]. The N concentration increase in the infiltrate also led to higher N leaching in the year of legume cultivation, despite lower rainfall infiltration in the legume crop. Compared to barley, legumes form much larger biomass during the growing season, which increases evapotranspiration leading to a corresponding reduction in rainfall infiltration in the summer and autumn seasons. The effect of legume residues on N leaching in temperate climates was effective 5–6 months after their incorporation. After the residue mineralisation was over, the differences between the effect of barley and legume residues on N leaching became insignificant. Analysis of the results of studies carried out in different European regions has shown that the effect of legumes on N leaching could be variable. On the one hand, legume-containing rotations increase N output, but this does not reduce N leaching in all cases. In some cases, due to higher N inputs into the soil with legumes, N leaching may increase compared to rotations without legumes [44,45]. 

The results of this research indicate that the climate warming that is taking place in Eastern Europe, by lengthening the duration of the warm period of the year, creates conditions for a longer mineralization of plant residues. As a result, with excessive moisture, the nitrogen losses from leaching from arable soils can increase. It is necessary to search for new agricultural practices that slow down the processes of mineralization of plant residues and the filtration of precipitation in the autumn.

## 4. Materials and Methods

### 4.1. Growth Conditions and Experimental Design

The experiment was carried out at the Vokė branch of the Institute of Agriculture, Lithuanian Research Centre for Agriculture and Forestry, located in the south-eastern part of Lithuania, which belongs to part of Central Europe (the East Baltic region). The experiment was carried out from March 2016 to February 2023. The influence of residues of legume crops grown in Lithuania (peas, beans and narrow-leaved lupins) on N accumulation and leaching in the soil was investigated. 

The climate in the region is moderate with a mean long-term (1991–2020) annual precipitation value of 695 mm, and an annual mean air temperature of 7.4 °C [25]. 

The effect of legumes on N_total_ and C_org_ concentration in soil and N leaching was studied in a *Fabaceae–Cereals* sequence. Three species of legumes—peas (*Pisum sativus* L.), fodder beans (*Faba beans*) and narrow-leaved lupines (*Lupinus angustifolium* L.)—were grown; spring barley (*Hordeum vulgare* L.) was grown as a control treatment. The *Fabaceae* were grown in 2016, 2018, 2020 and 2022. In the years following their cultivation (2017, 2019 and 2021), barley was sown and the effect of legume residues on yield and nitrogen leaching was studied.

The experiments were carried out in lysimetric stations (12 units) consisting of cylindrical concrete structures with a surface area of 1.75 m^2^, a depth of 0.90 m and a test soil layer of 0.60 m. Atmospheric precipitation percolating through the soil layer (infiltrate) was collected in 20 l receivers. After measuring the infiltrate, a sample was taken once a month to determine the nitrogen concentration (N-NO_3_ + N-NO_2_). 

The plants were grown on sandy loam Haplic Luvisol [46], which is very common in Lithuania. The humic layer thickness was 0.25 m. At the beginning of the experiment, the soil was slightly acidic (pH_KCl_ 5.0–5.2), moderately rich in plant-available phosphorus and potassium, with 1.18 g C_org_ kg^−1^ content. 

### 4.2. Agrotechniques for Growing Plants

All agrotechnical work on the lysimeters was carried out with hand tools, simulating conventional crop production technologies in the field. At the beginning of spring (second ten-day period of April) the soil was loosened to a depth of 8–10 cm. Before sowing, phosphorus and potassium fertiliser P_60_K_90_ (granular superphosphate and potassium chloride) was applied. Sowing took place at the end of the third ten-day period of April. Seed rates were 5.0 million ha^−1^ seeds for barley, 0.8 million ha^−1^ for fodder beans and 1.0 million ha^−1^ for peas and lupines. Appropriate pesticides were applied against plant diseases and pests during the growing season. 

In June–August, when aphids (*Aphididae* spp.) spread, the plants were treated with insecticide Proteus OD (Thiacloprid 100 g L^−1^, Deltamethrin 10 g L^−1^) at 0.6 L ha^−1^, 2–3 times per growing season. With the spread of the plant diseases *Ascochyta fabae Speg* and *Uromyces fabae d By*, the fungicide Ataka NT (mankozeb 750 g kg^−1^) was applied at 1.5 kg ha^−1^.

The grain yields were recorded at full maturity stage. During the year following legumes, barley was sown in the lysimeters. In order to determine the effect of the incorporated legume residues, no mineral fertilisation was applied.

### 4.3. Plant Yield Recording

The crops for grain and straw (barley, peas, beans and lupines) were harvested from the whole lysimeter area (1.75 m^2^). The plant’s biomass was cut about 2–3 cm above the ground. Plant leaf remains were collected from the soil surface and added to the cut biomass. The weight and moisture content of the grain and straw were determined and the dry matter yield of the grain and straw (DM g m^−2^) was calculated. The plant root mass was determined in each lysimeter by digging a soil monolith from a 0.5 m × 0.5 m area (0.25 m^2^). The root mass was determined in the 0–0.25 m layer, as ~98% of the root mass is formed there (unpublished data). The roots were gently washed of soil particles and dried at 60 °C. After harvesting, the plant stems were chopped to 4–5 cm length, spread on the surface of an appropriate lysimeter, and incorporated into the soil to a depth of 5–8 cm. In the second ten-day period of October, the soil in the lysimeters was dug to a depth of 20 cm. 

### 4.4. Measurement of Lysimetric Water

The rainfall infiltration rate (L m^−2^) was measured according to infiltration intensity, but at least once a month. Infiltration rates were calculated per month, per season (spring: 3rd–5th months, summer: 6th–8th months, autumn: 9th–11th months, winter: 12th–2nd months) and for the entire hydrological year (1 March–28 February of the following year). 

The precipitation amount was calculated according to the data of the Vilnius Meteorological Station, which is located 0.2 km away from the lysimetric facility site. The rainfall infiltration coefficient (Pk) was calculated using the formula:Pk (%) = Volume of percolating precipitation L m^−2^/precipitation amount L m^−2^
per year × 100

### 4.5. Methods of Chemical Analyses

The total nitrogen (N_total_ g kg^−1^) in the grain, stems and roots of plants was determined by the Kjeldahl method (Directive 72/199/EEC), C_org_—ISO 10694:1995.

For legumes, the biologically fixed nitrogen (BFN) content was calculated according to the coefficients proposed by Peoples et al. [30]—62% for peas and 74% for fodder beans and lupines.

For soil, the C_org_ was determined according to ISO 10694:1995 (determination of C_org_ after dry combustion—Dumas method) and total nitrogen by ISO 11261-1995.

The concentration of nitrogen (N-NO_3_ + N-NO_2_) in the water of the lysimeters was determined by LST EN ISO 13395:2000 (spectrometric detection). 

The weighted seasonal and annual nitrogen concentrations in the water were determined using the formula: N_avarage_ = ((N1 × VP1) + (N2 × VP2) + (N3 × VP3))/(VP1 + VP2 + VP3)
where: 

N_avarage_ is the weighted (monthly/seasonal) nitrogen concentration in the infiltrate mg N L^−1^; N1, N2, N3 are the average nitrogen concentrations mg N L^−1^ for a given month/season; and VP1, VP2, VP3 are the volumes of percolating precipitation in L m^−2^ per month/season.

### 4.6. Meteorological Conditions during Experiment

The air temperature during the experimental period (2016–2023) showed various deviations from SCN, however, among all the variations, a consistent upward trend could be identified for October and November. In January and February, the air temperature was mostly negative, with the exception of 2019 (Figure 4).

The annual precipitation varied from 529 mm in 2020 to 917 mm in 2017 (SCN 695 mm). The years 2017 and 2021 were rainy, with annual precipitation exceeding SCN by 14–32%, while 2018, 2019 and 2022 were drier years, with only 78–81% of SCN falling per year. In Lithuania, more precipitation falls in the summer period (68–84 mm per month on average), however, there was a large variation in average monthly precipitation over the entire study period (Figure 5). Very rainy months (monthly rainfall exceeding 100 mm) were June, July, August and October 2017, July 2018, August 2019 and 2021, and June and July 2022. Such hydrothermal conditions were favourable for rainfall infiltration. Depending on rainfall distribution over the months, its intensity and air temperature, the annual rainfall infiltration coefficient varied from 30.2% in 2020 to 63.0% in 2021. A regression correlation analysis showed that the amount of infiltrated rainfall had a stronger relationship (R^2^ = 0.42) with the amount of rainfall in spring periods. 

### 4.7. Statistical Analysis

The experimental data were evaluated by repeated measures analysis of variance (ANOVA), using the Fisher’s post-hoc test to carry out least significant difference between treatment. The probability level was set at 0.05. Duncan’s post-hoc test was used to assess the effect of legumes species on soil properties (C_org_ and N_total_ concentration) and the differences of plants biomass, post-harvest residues and root biomass, their chemical composition and nitrogen leaching. Average deviation (SE) values were used to estimate the deviations of soil chemical parameters from the mean values. A leaner regression analysis was used to reveal the relationship between nitrogen input with legume post-harvest residue biomass and N_total_ concentration in soil. 

## 5. Conclusions

On sandy loam soils, the highest amount of crop residues after harvesting the grain remains in the field under narrow-leaved lupines (*p* < 0.05), and less remains after beans and peas. The main DM content is incorporated with the above-ground part of the plant. The root biomass accounts for 4.6–15.8% of total post-harvest residues. An amount of 4.01–8.03 g N m^−2^ is returned to the soil with legume residues. The highest N return is associated with lupine residues (*p* < 0.05). The N concentration in the above-ground part and roots varies from 5.1 to 17.6 g N kg^−1^ depending on the species and plant part. The C/N ratio in legume residues is favourable enough for mineralisation: C/N = 31.4–49.8 in the above-ground part, and C/N = 25.5–43.6 in the roots. The incorporation of legume residues increased the N_total_ concentration in the soil. Significantly higher N concentrations were found in the soil with the lupines and beans, while the effect of the pea residues did not differ significantly from the effect of barley residues on this indicator. The legume residues had a similar effect on C_org_ accumulation in soil. 

Climate warming (an increase in the duration of the warm period and a reduction in the duration of the cold period of the year) creates favourable conditions for the mineralization of nitrogen-rich legume residues in the soil and N leaching. Compared to barley residues, the N-NO_3_ concentration in the lysimeter water after the incorporation of legume residues increased on average by 14–70% in the autumn and 14–86% in the winter seasons. As a result, N leaching losses significantly increased (+24.7–33.2%, *p* < 0.05) during the years of grain legume cultivation. The N leaching losses were correlated with the amount of N incorporated with residues, and, therefore, the lupine residues increased N leaching more than the bean and pea residues. In the following year, the effect of legume residues on nitrate concentrations and N leaching was reduced to be similar to that of the barley residues.

The impact of legume crops on nitrogen leaching from the soil should be taken into account when assessing the factors affecting the extent of pollution of surface groundwater with nitrogen compounds, since with a significant increase in their area, this effect in certain regions can become significant.

## Figures and Tables

**Figure 1 plants-12-02478-f001:**
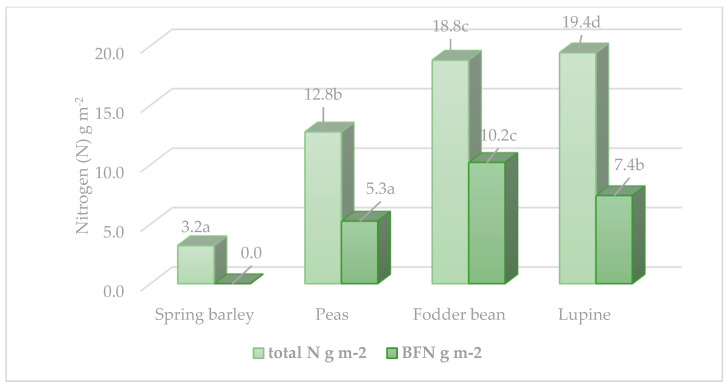
Accumulation of total and biofixed nitrogen (g N m^−2^) in plant biomass. Note: different letters in columns indicate a significant difference between plant species (*p* < 0.05).

**Figure 2 plants-12-02478-f002:**
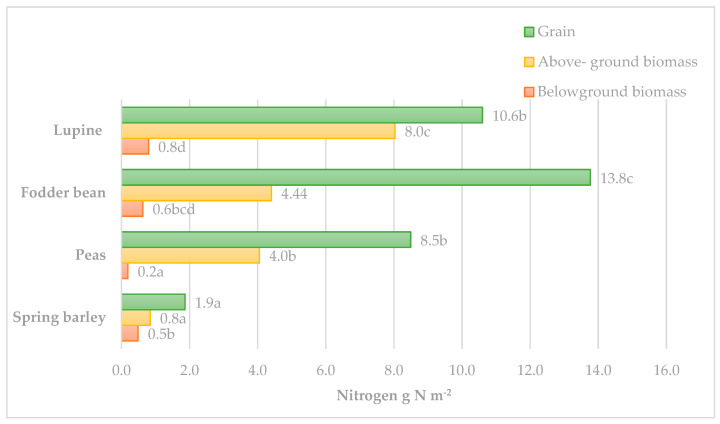
Nitrogen accumulation (N g m^−2^) in individual plant parts. Note: different letters in columns indicate a significant difference between plant species (*p* < 0.05).

**Figure 3 plants-12-02478-f003:**
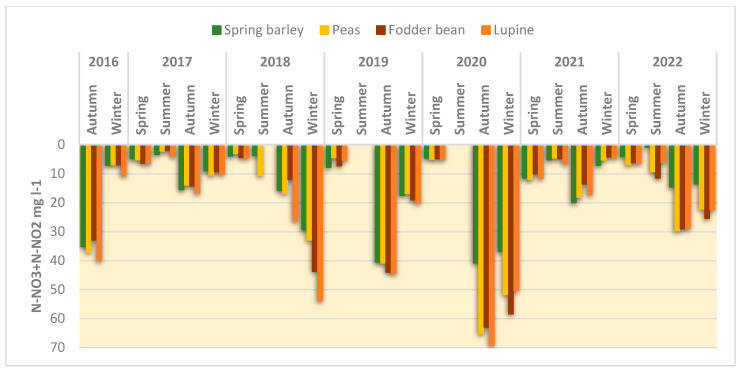
Nitrogen (NO_2_-N + NO_3_-N mg L^−1^) concentrations in lysimeter water, data of March 2016–February 2023.

**Figure 4 plants-12-02478-f004:**
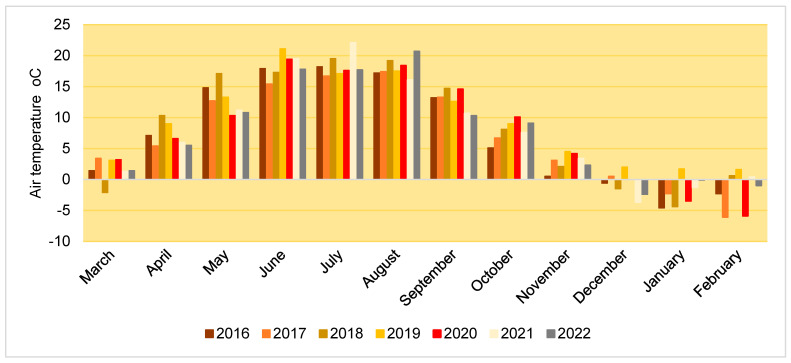
The air temperature (°C) during the experimental period (2016–2023).

**Figure 5 plants-12-02478-f005:**
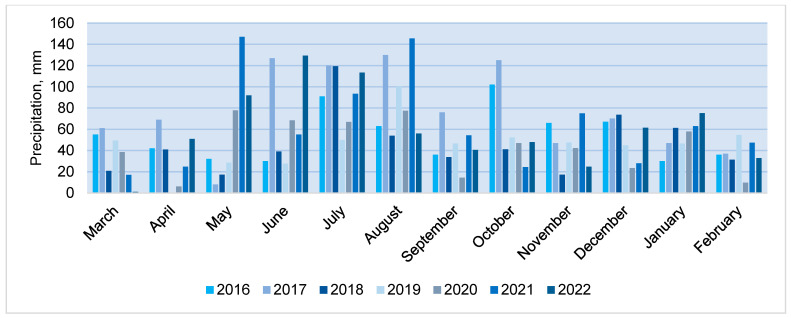
The precipitation amount (mm) during the experimental period (2016–2023).

**Table 1 plants-12-02478-t001:** Biomass yields of individual morphological plant parts in g DM m^−2^ (average data for 2016, 2018, 2020 and 2022).

Plant Species	Total Plant Biomass	Including			
		Grain Yield	Above-Ground Biomass	Below-Ground Biomass	Above- and Below-Ground Biomass
Spring barley	370.87	157.4a	158.3a	55.1b	213.4a
Peas	638.0bc	197.8b	419.9c	20.3a	440.2b
Fodder bean	646.6bc	274.8c	313.0d	58.9bcd	371.8b
Narrow-leaved lupine	690.3c	146.1a	478.3c	65.9d	544.1c

Note: different letters in columns indicate a significant difference between plant species (*p* < 0.05).

**Table 2 plants-12-02478-t002:** Nitrogen concentrations (g N kg^−1^) in individual plant parts.

Plant Species	Grain	Above-Ground Biomass	Below-Ground Biomass
Spring barley	13.5 ± 1.53a	5.1 ± 0.067a	7.0 ± 0.185a
Peas	34.0 ± 1.78d	9.0 ± 0.158abc	17.6 ± 0.288b
Fodder bean	42.5 ± 0.62c	11.3 ± 0.302c	8.9 ± 0.372a
Narrow-leaved lupine	44.1 ± 1.64c	8.4 ± 0.120abc	10.8 ± 0.278a

Note: different letters in columns indicate a significant difference between plant species (*p* < 0.05).

**Table 3 plants-12-02478-t003:** Carbon concentrations in above- and below-ground parts of plants.

Plant Species	C %		C/N	
	Above-Ground Biomass	Below-Ground Biomass	Above-Ground Biomass	Below-Ground Biomass
Spring barley	38.9 ± 1.56d	40.5 ± 1.66a	76.4	57.8
Peas	42.6 ± 2.63c	44.9 ± 1.34c	47.4	25.5
Fodder bean	35.5 ± 1.31a	38.8 ± 1.32a	31.4	43.6
Narrow-leaved lupine	41.8 ± 2.00c	43.5 ± 2.29c	49.8	40.3

Note: different letters in columns indicate a significant difference between plant species (*p* < 0.05).

**Table 4 plants-12-02478-t004:** C_org_ and N_total_ concentrations (g kg^−1^) in soil at the end of the experiment (2022).

Plant Species	C_org_	N_total_
Spring barley	11.1a	0.84a
Peas	12.6ab	0.90ab
Fodder bean	14.4d	0.96bcd
Narrow-leaved lupine	13.6bcd	1.00d

Note: different letters in columns indicate a significant difference between plant species (*p* < 0.05).

**Table 5 plants-12-02478-t005:** Rainfall infiltration and nitrogen leaching in sandy loam soils.

Plant Species	Infiltration Rate per Hydrological Year L m^−2^		Nitrogen Leaching Losses g N m^−2^	
	Legume Cultivation Year	Barley Cultivation Year	Legume Cultivation Year	Barley Cultivation Year
Barley/barley	263.0b	406.4b	4.58a	5.24ab
Peas/barley	243.1a	396.8ab	5.71bc	4.87ab
Fodder beans/barley	231.3a	394.6ab	5.84bc	4.76ab
Lupines/barley	236.3a	396.9ab	6.10c	5.28b

Note: different letters in columns indicate a significant difference between plant species (*p* < 0.05).

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
