# Peer review of "Organic Carbon, Nitrogen Accumulation and Nitrogen Leaching as Affected by Legume Crop Residues on Sandy Loam in the Eastern Baltic Region"

_plants, 2023, doi:10.3390/plants12132478_

Round 1
Reviewer 1 Report
This manuscript describes a study on leguminous plants promoting soil carbon and nitrogen accumulation and increasing the risk of nitrogen leaching. This study has practical significance and reference value, but there are shortcomings in the manuscript writing, and the acceptance was considered again after major revisions.
The title mentioned nitrogen cycle, but all test variables shown in the manuscript could not well characterize and reflect the nitrogen cycle process of soil or soil-crop system, so it is suggested to modify the title.
Please briefly introduce the experimental setup in the abstract, otherwise when reading Line 26 may not understand why it should be compared to barley.
Please provide the reasons for choosing these three leguminous plants. And the results include a comparison of the differences in the three leguminous plants themselves or their environmental effects. What is the significance of whether there are differences between them.
The author stated that the objective of this study was to investigate the effect of legume residues returning to field under warming climate conditions, but the experimental design did not reflect warming, and the experimental results did not include an analysis of temperature. I don't understand how to achieve the author's research objectives.
In fact, the results and phenomena of this experiment are all predictable, that is, returning leguminous plant residues to the field has a positive effect on soil carbon and nitrogen accumulation and nitrogen leaching. So how can the innovation of this study be reflected in this manuscript? Or is it possible to make management recommendations for agricultural production based on the research results?
No comments
Author Response
Dear Reviewers,
Thank you for the comments. Please accept the answers.

Reviewer 2 Report
The article presents useful information on legume residues management and has consistent experimental scope, although only one kind of soil was investigated. The manuscript is well written and requires minor editions.
In Abstract, I would prefer to reduce introductory phrases and to introduce methodological statements on plant growth and harvest to guide the reader; however it is a matter of choice. For instance, it is not clear that there was a barley – legume rotation, and that the residues were produced after grain harvested. Additionally, it is a long run study that should be stated.
The Methodology describes that legumes were harvested at grain maturity. Most legume species have strong leaf abscission near grain maturity and most leaves are hard to retrieve. I think that this matter should be commented in the methodology: most leaves were still attached to plants at maturity and were recovered, or fallen leaves over the soil were (or not) collected?
In Tables and Figures, the footnote should quote the test for mean comparison. Particularly in Figure 1, the footnote should state that the letters compare species (nor plant parts). In Table 1, I could not understand the column “Grain/above and below ground biomass”; this was not commented in the text.
I am not sure on the adequacy of presenting data of biomass yield in g m-2 (Table 1) but data f N accumulation in kg ha-1 (Figures 1 and 2). I would prefer everything in g m-2.
Line 197 and so on: L in capital letter.
The Conclusions are long and repeat many results. I would try to synthetize it but it is again a matter of style.
Author Response

(The authors gave the same response as above.)

Reviewer 3 Report
line 333: please add the specification and calendarization of pesticide treatments
needs a critical reading by a native english speaker, minor corrections needed
Author Response

(The authors gave the same response as above.)

Round 2
Reviewer 1 Report
The P value for statistical analysis can be in both uppercase and lowercase forms, please unify.
Why is the abbreviation for biologically fixed nitrogen not BFN?
Author Response
Please accept the answer.
